# Learning to Specialize with Knowledge Distillation for Visual Question Answering

**Jonghwan Mun**[1,3]     **Kimin Lee**[2]     **Jinwoo Shin**[2]     **Bohyung Han**[3]

[1]Computer Vision Lab., POSTECH, Pohang, Korea
[2]Algorithmic Intelligence Lab., KAIST, Daejeon, Korea
[3]Computer Vision Lab., ASRI, Seoul National University, Seoul, Korea

[1]`choco1916@postech.ac.kr`  [2]`{kiminlee,jinwoos}@kaist.ac.kr`  [3]`bhhan@snu.ac.kr`

## Abstract

Visual Question Answering (VQA) is a notoriously challenging problem because it involves various heterogeneous tasks defined by questions within a unified framework. Learning specialized models for individual types of tasks is intuitively attracting but surprisingly difficult; it is not straightforward to outperform naïve independent ensemble approach. We present a principled algorithm to learn specialized models with knowledge distillation under a multiple choice learning (MCL) framework, where training examples are assigned dynamically to a subset of models for updating network parameters. The assigned and non-assigned models are learned to predict ground-truth answers and imitate their own base models before specialization, respectively. Our approach alleviates the limitation of data deficiency in existing MCL frameworks, and allows each model to learn its own specialized expertise without forgetting general knowledge. The proposed framework is model-agnostic and applicable to any tasks other than VQA, *e.g.*, image classification with a large number of labels but few per-class examples, which is known to be difficult under existing MCL schemes. Our experimental results indeed demonstrate that our method outperforms other baselines for VQA and image classification.

## 1   Introduction

Visual Question Answering (VQA) [9] is a task to find an answer for a question about an input image. This is an extremely challenging problem because VQA models deal with various recognition tasks at the same time within a unified framework, which requires to understand local and global context of an image as well as a question. A VQA model thus should have diverse reasoning capabilities to capture appropriate information from input images and questions. Despite such challenges, recent approaches [4, 8, 16, 27, 31, 33, 34] show impressive performance by leveraging advance of deep neural networks and emergence of large-scale datasets [9, 14].

Although VQA is composed of various tasks defined by questions, existing algorithms typically train a universal model generalized for all possible questions as depicted in Figure 1(a). This is partly because designing and learning specialized models on each task is difficult by itself and it is not straightforward to develop an algorithm assigning tasks to a subset of models in a principled way. In practice, it is challenging to show improved performance by model specialization compared to naïve independent ensemble. This paper tackles how to associate models with individual types of tasks and how to learn the specialized models effectively as illustrated in Figure 1(b).

Recently, Multiple Choice Learning (MCL) [10, 19, 21] has been investigated as an elegant framework to learn specialized models for recognition. In MCL, examples are typically assigned to a subset of models with the highest accuracy, thus each model is expected to be specialized to certain types

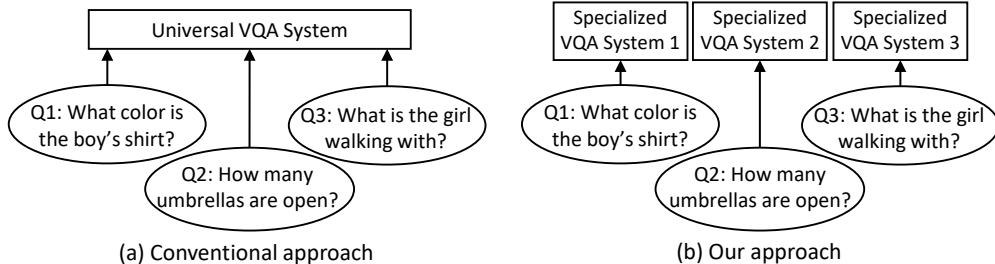

(a) Conventional approach          (b) Our approach

Figure 1: Comparison between the conventional approach and ours for VQA. In our approach, we are interested in learning specialized models for certain questions that require different visual reasoning.

of examples. Our intuition is that the specialized models have potential to outperform the models generalized on all tasks since the models trained by MCL achieve higher oracle accuracy; at least one of the models predicts correctly for each example. This suggests that approaches to learning specialized models would be a promising direction to beat an ensemble of universal models in VQA.

Unfortunately, direct use of MCL turns out to be ineffective in VQA, *i.e.*, the models trained with the existing MCL schemes typically are outperformed by naïve ensembles. This is mainly because models trained by MCL inherently suffer from the problem of data deficiency due to hard assignments of tasks to particular models; each model can see only a subset of training dataset, which results in weak generalization accuracy. In addition to such data deficiency issue, MCL loses the opportunity to learn general knowledge from the examples assigned to other models. Note that VQA models tend to learn some compositional information observed in all training examples. For example, given two questions 'what is the color of umbrella?' and 'how many people are wearing glasses?' in a training set, a model specialized to only one of the two questions may have troubles to answer a question like 'what is the color of glasses?' Therefore, one has to specialize models while being aware of general knowledge shared across diverse examples. The issue has not been addressed in the MCL approaches as they mainly focus on simple image classification tasks with at most 10 labels [10, 19, 21].

**Contribution**    To overcome such challenges, we propose a novel and principled algorithm leveraging multiple choice learning and knowledge distillation. The main difference with respect to the standard MCL framework is two-fold: oracle loss definition and example assignment policy. We first learn multiple base models independently on the whole training dataset, and assign each example based on the scores from the base models. During training procedure, we specialize a subset of models to the dynamically assigned examples given by the current configurations of models while the rest of models imitate predictions of base models for the examples. This strategy of knowledge distillation with non-assigned examples alleviates the data deficiency problem of MCL and is effective to learn rich compositional information across examples. The proposed algorithm shows meaningful performance improvement over the naïve ensemble and variants of MCL on VQA. Furthermore, due to its model-agnostic and generic property, it is straightforward to apply the proposed method to a variety tasks.

The main contribution of this paper is summarized as follows:

- We propose a novel model-agnostic ensemble learning algorithm, referred to as Multiple Choice Learning with Knowledge Distillation (MCL-KD), which learns models to be specialized to a subset of tasks. In particular, we introduce a new oracle loss by incorporating the concept of knowledge distillation into MCL, which facilitates to handle data deficiency issue in MCL effectively and learn shared representations from whole training data.

- The proposed algorithm is applied to existing VQA models and consistently improves performance compared to independent ensembles and existing MCL-based approaches.

- We present that our framework also works well in challenging image classification tasks with many labels but few per-class examples, in which other MCL variants perform poorly.

**Related works**    The main research stream of VQA is to learn end-to-end deep neural network models that answer all types of questions in a unified framework. For the purpose, VQA models

incorporate various techniques to concentrate only on the tasks specified by questions, including attention mechanism [26, 27, 31, 33, 34], multimodal fusion schemes [4, 8, 16], adaptive networks [28] or modular networks [2, 3, 13]. Although these approaches show capability to adapt models to particular tasks characterized by input images and questions, it is not straightforward to handle various heterogeneous tasks in a single model and understand internal operations of VQA models.

Use of multiple models for VQA is a natural direction because ensemble of multiple models is a common practice to improve performance in deep learning and proper allocations of tasks to a subset of the models may lead to better generalization by reducing problem complexity. Traditional independent ensemble (IE) [7], which trains models independently with random initialization, is known as a reasonable option but is far from the approaches to learn task-specific models. A more sophisticated ensemble method based on MCL [10] specializes ensemble members on a subset of data and encourages individual models to produce diverse and reasonable outputs; it minimizes the so-called *oracle loss* and focuses on the most accurate prediction. However, due to overconfidence issue [19, 20] of deep neural networks (DNNs), it is not straightforward to select appropriate models from ensemble members. The confident multiple choice learning (CMCL) [19] alleviates this issue by introducing a loss term that minimizes the Kullback-Leibler (KL) divergence between the predictive distribution of each non-specialized model and a uniform distribution. However, it suffers from the data deficiency issue for complex tasks such as VQA, as we mentioned earlier. We overcome the limitation by incorporating knowledge distillation [12].

Knowledege distillation is achieved by training a network to mimic the activations of intermediate layers [30], attention maps [36] or output distributions [12] of large networks (*i.e.*, teacher network). The knowledge distillation technique is widely used to learn compact and fast small models (*i.e.*, student network) in many practical applications [5, 22, 25]. In addition to learning compact models, the concept of knowledge distillation is used in other tasks [6, 23, 29]. Chen *et al.* [6] proposes a system building large deep neural network models by transferring knowledge from small networks trained beforehand while Li *et al.* [23] employs idea for continual learning to preserve knowledge from previously learned tasks. Noroozi *et al.* [29] boosts performances in a self-supervised learning framework by adopting knowledge distillation pipeline instead of fine-tuning when passing learned feature information to target tasks. In contrast to the prior works, our novelty lies in utilizing knowledge distillation for balancing generalization and specialization of ensemble models.

## 2 Background on Multiple Choice Learning

This section introduces the main idea of multiple choice learning, and compares its two variants, original and confident multiple choice learning.

### 2.1 Multiple Choice Learning

The objective of MCL [10] is to minimize the oracle loss, *i.e.*, making at least one of $M$ models predict the correct answer. Formally, denote a training dataset by $\mathcal{D} = \{(x_1, y_1), (x_2, y_2), ..., (x_N, y_N)\}$, where $N$ is the number of training examples and $x$ and $y$ are input and ground-truth output, respectively. Given multiple predictive distributions $P$ from $M$ models, the oracle loss is defined as

$$\mathcal{L}_{\text{MCL}}(\mathcal{D}) = \sum_{i=1}^{N} \min_{m} \ell_{\text{task}}(y_i, P(y|x_i; \theta_m)), \tag{1}$$

where $\theta_m$ and $\ell_{\text{task}}(\cdot, \cdot)$ denote the $m$-th model parameter and a task specific loss function, respectively. The oracle loss optimizes the most accurate model for individual input examples $x_i$, driving each model to become a specialist on a subset of questions.

Since the minimization function is not a continuous function, the oracle loss is relaxed to the following integer programming problem:

$$\mathcal{L}_{\text{MCL}}(\mathcal{D}) = \sum_{i=1}^{N} \sum_{m=1}^{M} v_{i,m} \ell_{\text{task}}(y_i, P(y|x_i; \theta_m)) \quad \text{subject to} \quad \sum_{m=1}^{M} v_{i,m} = k, \tag{2}$$

where $v_{i,m} \in \{0, 1\}$ is an indicator variable for the assignment of $x_i$ to the $m$-th model, and $k$ $(= 1, \ldots, M)$ is the number of specialized models per example. Note that if $k = M$, MCL is identical to independent ensemble.

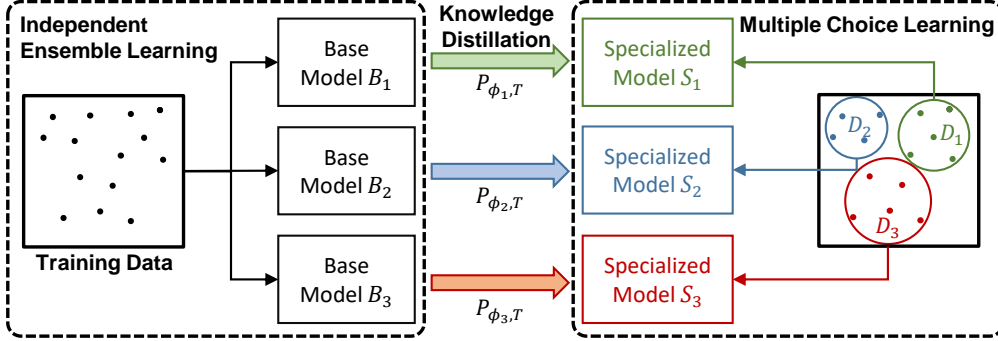

Figure 2: Overall framework of our multiple choice learning with knowledge distillation.

## 2.2 Confident Multiple Choice Learning

Although the objective of MCL [10] is appropriate to generate diverse outputs, it suffers from overconfidence issue that results in failure of selecting specialized models during inference. To address this issue, Lee *et al.* [19] proposed Confident Multiple Choice Learning (CMCL) based on a confident oracle loss:

$$\mathcal{L}_{\text{CMCL}}(\mathcal{D}) = \sum_{i=1}^{N} \sum_{m=1}^{M} v_{i,m} \ell_{\text{task}}(y_i, P(y|x_i; \theta_m)) + \beta(1 - v_{i,m}) D_{KL}(\mathcal{U}(y)||P(y|x_i; \theta_m))$$

$$\text{subject to} \quad \sum_{m=1}^{M} v_{i,m} = k, \quad v_{i,m} \in \{0, 1\}, \tag{3}$$

where $\beta$ is a weight for the losses of non-specialized models, $D_{KL}$ is KL divergence and $\mathcal{U}(y)$ denotes the uniform distribution. Compared to oracle loss in MCL, the confident oracle loss further regularizes for non-specialized models to be less confident by minimizing the KL divergence between the predictive and uniform distributions. Although the oracle loss in CMCL is well-designed to learn specialized models, its performance is not impressive particularly in complex tasks such as VQA because the loss function enforces individual models to disregard unassigned training examples completely. To tackle the limitations of the existing MCL techniques, we present a more principled oracle loss in the next section.

## 3 Multiple Choice Learning with Knowledge Distillation

Although MCL and CMCL show potential to achieve competitive performance compared to independent ensembles, model specialization on a subset of training examples suffers from weak generalization power of each model, often resulting in degraded accuracy. We believe that this is mainly due to data deficiency; specialization to a subset of training data reduces the number of observed examples for each model.

To address the issue, we design a novel oracle loss that makes models specialized to a subset of training examples while encouraging them to maintain *common sense* in training data (*e.g.*, the concept of attributes, objects and etc). This objective is achieved by multiple choice learning with knowledge distillation. Our overall learning framework is depicted in Figure 2, which is composed of two steps. Our algorithm first learns $M$ base models, which are generalists trained on the whole training dataset independently. Then, we specialize a subset of models to each example while the rest of models are trained to be at least as good as the corresponding base models on the example.

Another motivation of our work starts from a natural question that forcing uniform distribution is optimal choice for relaxing the overconfidence issue of MCL. We indeed found that CMCL is not effective in more complex tasks such as VQA. This fact motivates our approach of developing a new loss function, which utilizes the knowledge distillation.

### 3.1 Multiple Choice Learning with Knowledge Distillation

We propose a novel multiple choice learning framework, Multiple Choice Learning with Knowledge Distillation (MCL-KD). Given a training dataset $\mathcal{D} = \{(x_1, y_1), (x_2, y_2), ..., (x_N, y_N)\}$ and $M$ independently trained base models with fixed model parameters $\phi_m$, we train $M$ models parametrized by $\theta_m$ in the proposed MCL-KD framework using the following oracle loss:

$$\mathcal{L}_{\text{MCL-KD}}(\mathcal{D}) \tag{4}$$
$$= \sum_{i=1}^{N} \sum_{m=1}^{M} v_{i,m} \ell_{\text{task}}(y_i, P(y|x_i; \theta_m)) + \beta(1 - v_{i,m})\ell_{\text{KD}}\left(P(y|x_i; \phi_m, T), P(y|x_i; \theta_m, T)\right),$$

$$\text{subject to} \quad \sum_{m=1}^{M} v_{i,m} = k, \quad v_{i,m} \in \{0, 1\},$$

where $\beta > 0$ is a hyper-parameter to balance the two loss terms, and $T > 0$ is a temperature scaling parameter of the softmax function. Here, we employ a knowledge distillation loss between the $m$-th base model and the corresponding specialized model $\ell_{\text{KD}}(\cdot)$, which is formally given by

$$\ell_{\text{KD}}\left(P(y|x_i; \phi_m, T), P(y|x_i; \theta_m, T)\right) = D_{KL}\left(P(y|x_i; \phi_m, T)||P(y|x_i; \theta_m, T)\right), \tag{5}$$

where $P(y|x; \theta, T) = \frac{\exp(f_y(x;\theta)/T)}{\sum_{y'} \exp(f_{y'}(x;\theta)/T)}$ is a calibrated softmax distribution, and $f(\cdot)$ denotes a logit of deep models.

In our oracle loss, specialized models are learned to predict the ground-truth answers while non-specialized ones are trained to preserve the representations of the corresponding base models. We believe that this is a reasonable choice because knowledge distillation is known for an effective technique for fast optimization, transfer learning, and learning without forgetting [23, 35]. Note that, contrary to CMCL, the knowledge distillation loss in our framework provides opportunity to learn from non-assigned training examples and makes non-specialized models as competitive as the corresponding base models for those examples.

Training procedure of MCL-KD is as follows. We first learn M base models using the whole training data independently. Given the number of specialized models per example denoted by $k$, a binary assignment vector by $v_i = (v_{i,1}, v_{i,2}, ..., v_{i,M})$ indicates which models are specialized for $x_i$. Then, $v_i$ and $\theta_m$ are optimized by the following iterative procedure:

1. **Fix $\theta_m$ and optimize for $v_i$.**
   Let us denote the collection of all possible assignments vectors by $\mathcal{A}_{k,M}$. Then, based on the current model parameters $\theta_m$, the assignment vector $v_i$ is determined to achieve the lowest $\mathcal{L}_{\text{MCL-KD}}$, which is given by

   $$v_i = \underset{v_i' \in \mathcal{A}_{k,M}}{\arg\min} \mathcal{L}_{\text{MCL-KD}}(x_i, y_i, v_i')$$
   $$= \underset{v_i' \in \mathcal{A}_{k,M}}{\arg\min} \sum_{m=1}^{M} v_{i,m}' \ell_{\text{task}}(y_i, P_{\theta_m}(y|x_i)) + \beta(1 - v_{i,m}')\ell_{\text{KD}}\left(P_{\phi_m, T}(y|x_i), P_{\theta_m, T}(y|x_i)\right).$$

2. **Fix $v_i$ and optimize for $\theta_m$.**
   Given assignment vector $v_i$, each model is trained to minimize the task loss for assigned examples and the knowledge distillation loss for non-assigned examples.

These two optimization steps are repeated until convergence. For computational efficiency, Lee *et al.* [21] proposed a stochastic algorithm that the model assignment step and optimizing models are performed inside a batch. That is, examples are assigned to a model with the lowest oracle loss and models are updated without assignment convergence in a batch. Because the simple stochastic algorithm is computationally efficient and works well in practice, we also employ this option for optimization of the proposed oracle loss.

Table 1: Classification accuracy (%) on CLEVR validation set with varying the number of specialized models ($k$). The bold-faced numbers mean the best algorithm for each $k$ in top1 accuracy.

| Answering Network | $k$ | Single Top1 | IE Top1 | IE Oracle | MCL Top1 | MCL Oracle | CMCL Top1 | CMCL Oracle | MCL-KD Top1 | MCL-KD Oracle |
|---|---|---|---|---|---|---|---|---|---|---|
| MLP | 1 | | | | 41.31 | 98.92 | 59.12 | 63.76 | **60.22** | 80.75 |
| | 2 | 58.40 | 60.10 | 80.73 | 48.94 | 97.57 | 60.27 | 76.00 | **60.38** | 81.20 |
| | 3 | | | | 58.63 | 95.67 | 60.49 | 82.67 | **60.89** | 81.86 |
| SAN | 1 | | | | 42.19 | 98.67 | 83.99 | 91.55 | **85.98** | 95.38 |
| | 2 | 82.30 | 85.23 | 94.93 | 58.39 | 98.62 | 84.83 | 96.64 | **87.02** | 95.78 |
| | 3 | | | | 83.73 | 98.62 | 86.18 | 96.26 | **88.16** | 96.12 |

Table 2: Classification accuracy (%) on VQA v2.0 validation set at $k = 3$. The bold-faced number means the best algorithm in top1 accuracy.

| Single Top1 | IE Top1 | IE Oracle | MCL Top1 | MCL Oracle | CMCL Top1 | CMCL Oracle | MCL-KD Top1 | MCL-KD Oracle |
|---|---|---|---|---|---|---|---|---|
| 63.42 | 65.27 | 76.23 | 64.94 | 78.15 | 64.99 | 73.34 | **65.67** | 76.95 |

## 3.2 Application to Visual Question Answering

On VQA, given the input data of an image and a question $x_i = (I_i, q_i)$ and its corresponding answer $y_i$, an answering network is trained to minimize a negative log-likelihood of prediction:

$$\ell_{\text{VQA}}(y_i, P(y|x_i; \theta)) = -\log P(y_i|x_i; \theta) = -\log P(y_i|I_i, q_i; \theta). \quad (6)$$

In our algorithm, specialized answering networks with knowledge distillation are trained with the loss function in Eq. (4) with $\ell_{\text{task}}$ replaced by $\ell_{\text{VQA}}$. The loss functions of all other compared models such as IE, MCL, and CMCL are obtained by the same manner. After learning specialized models, we feedforward a testing example for each model and the final answer is obtained by the average of all prediction scores after softmax, which is given by

$$\arg\max_y \; \frac{1}{M} \sum_{m=1}^{M} P(y|I, q; \theta_m), \quad (7)$$

where $\theta_m$ denotes the parameters of the $m$-th specialized model.

# 4 Experimental Results

## 4.1 Visual Question Answering

**Dataset** We employ CLEVR and VQA v2.0 datasets to validate our algorithm. CLEVR [14] is constructed for an analysis of various aspects of visual reasoning such as attribute identification, counting, comparison, spatial relationship, and logical operations. The dataset is composed of 70,000 training images with 699,989 questions and 15,000 validation images with 149,991 questions, where each question is associated with a single unique answer. The vocabulary sizes of question and answer are 84 and 28, respectively. VQA v2.0 [9] is a very popular dataset based on images collected from MSCOCO [24]. To handle dataset bias issues found in v1.0, it contains two images with different answers for each question. This dataset consists of 443,757 and 214,354 questions for train and validation, respectively, where each question has 10 candidate answers.

**Experimental setup** For CLEVR dataset, we adopt two models as our answering networks: a simple MLP-based model with 2 hidden layers of 1,024 units after an image and question fusion layer, and a well-known stacked attention network (SAN) [15]. We extract `conv4` features from input images of 224×224 using ResNet-101 [11], which results in 1024×14×14 image representations. We also apply additional residual blocks on top of the extracted `conv4` features to adapt the image representations to a target dataset. All models are optimized using ADAM [17] with fixed learning rate of 0.0005 and batch size of 64 while the parameters of ResNet-101 are fixed. We set $\beta$ and $T$ in Eq. 4 to 50 and 0.1, respectively, based on our empirical observations. For VQA v2.0 dataset, we adopt the bottom-up and top-down attention model [1], which is the winner of 2017 VQA challenge.

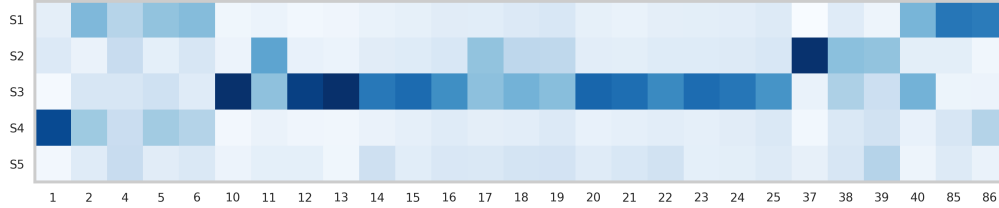

Figure 3: Visualization of the number of training examples assigned to each SAN model [15] of MCL-KD at $k = 1$ in CLEVR dataset. The numbers in $x$-axis denote indices of question families, and the number of assigned examples are normalized across each question family (each column).

Table 3: Question examples of a question family in which SAN models in Figure 3 are dominantly specialized. Questions in different question family ask different semantics in images. For examples, questions in #1 are about comparison with counting two objects while those in #37 requires comparison of size between objects.

| Question family | Question examples |
|---|---|
| #1 | Are there the same number of red balls and cyan balls? |
| | Are there an equal number of metallic spheres and brown cylinders? |
| | Is the number of yellow things the same as the number of tiny cyan blocks? |
| | Are there the same number of blue cylinders and big blocks? |
| #13 | Does the shiny object in front of the purple matte block have the same size as the small metallic cylinder? |
| | There is a object that is behind the blue object; does it have the same size as the green cylinder? |
| #37 | Are there any other things that have the same size as the brown shiny sphere? |
| | Is there any other thing that is the same size as the rubber cube? |
| | Is there any other thing of the same size as the cube? |
| | Are there any other things that have the same size as the cyan metallic cylinder? |
| #85 | How many tiny blue cubes are there? |
| | How many matte cubes are there? |
| | What number of big red objects are there? |
| | What number of tiny yellow metal cylinders are there? |

We use the publicly available implementation[1] and leave all parameters unchanged except batch size, which is changed from 512 to 256 due to memory limitation. We set $\beta$ and $T$ to 5,000 and 0.5, respectively.

For fair comparison, we initialize the networks of all algorithms (MCL, CMCL and MCL-KD) in the same way using the base models trained independently. According to our observation, this strategy generally achieves higher performance than learning from scratch for all methods. For evaluation, we measure both top-1 and oracle accuracy. The top-1 accuracy is computed by the ratio of correctly predicted examples identified by the average output distribution of ensemble members. The oracle accuracy measures whether at least one of the models predicts the correct answers for input image and question pairs. Generally speaking, higher oracle accuracy implies that trained models are specialized more to subsets of data.

**Results** We compare our algorithm denoted by MCL-KD with three baselines—IE, MCL and CMCL. We train 5 models while varying the number of specialized models, $k$, for MCL, CMCL and MCL-KD. We test performance of MCL-based models when $k = 1, 2, 3$.

Table 1 summarizes the results on the validation set of CLEVR dataset. It is noticeable that top-1 accuracies in all three MCL-based methods are getting higher with larger $k$. This is partly because increasing $k$ is effective to alleviate data deficiency issue and improve quality of ensemble predictions. MCL is the best in terms of oracle accuracy, but its top-1 accuracy is not satisfactory due to overconfidence issue. CMCL is substantially better than MCL, but still not sufficient to achieve clear accuracy improvement with respect to IE. On the contrary, MCL-KD consistently outperforms IE, MCL, and CMCL regardless of $k$. This implies that applying knowledge distillation loss for

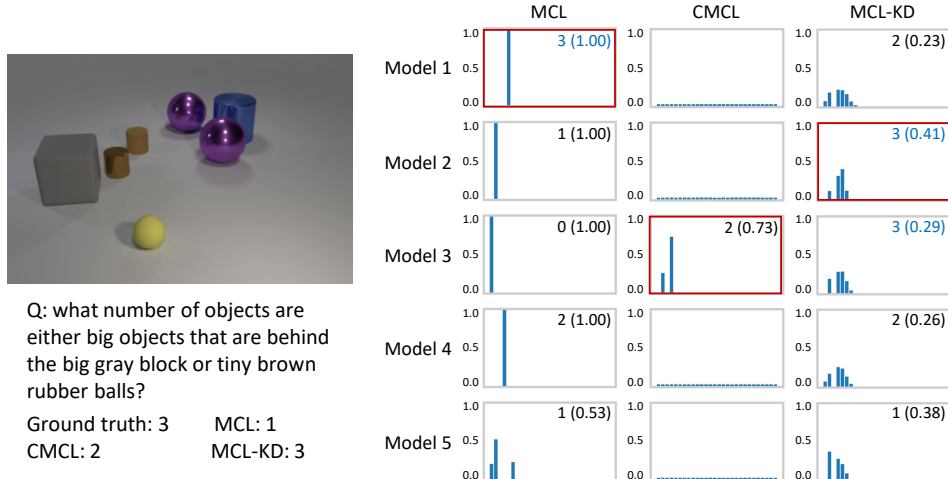

Q: what number of objects are either big objects that are behind the big gray block or tiny brown rubber balls?

Ground truth: 3     MCL: 1
CMCL: 2              MCL-KD: 3

Figure 4: Visualization of prediction distributions given by SAN models of MCL, CMCL and MCL-KD algorithms in CLEVR dataset when $k = 1$. The number at the upper-right corner of each box means the predicted label and its probability of each model, where correct predictions are marked in blue. Red boxes denote the specialized models.

non-specialized models is important to balance specialization and generalization of ensemble models in visual question answering. We also observe that the accuracy improvement compared to IE in MLP is lower than in SAN. This is probably because of the following two reasons: 1) reduced benefit of ensemble because the architecture of the MLP is simpler and the learned models are more correlated, and 2) degraded knowledge distillation performance since single model accuracy of the MLP is lower thus there is little knowledge to be distilled compared to stronger models.

Table 2 presents the results on the validation set of VQA v2.0 dataset at $k = 3$. VQA v2.0 dataset is more complex and challenging since it is a real dataset and contains more diverse question and answer pairs than CLEVR dataset. Nevertheless, MCL-KD still achieves accuracy gain with respect to IE while both MCL and CMCL are not effective enough to improve accuracy by model specialization.

**Analysis on CLEVR**   CLEVR dataset divide the whole dataset into 90 question families depending on their requirements of visual reasoning, and examples with a kind of questions belong to the same family. Using the information, we analyze SAN models in MCL-KD ($k = 1$) on CLEVR dataset and illustrate how question families are associated with individual models. Figure 3 presents model specialization tendency of MCL-KD over the question families. It is interesting to see that 4 models (S1, S2, S3, and S4) out of 5 are specialized to the unique subsets of question families while S5 mimics its corresponding base model since only a small number of examples are assigned to S5 and distribution of question family assignment is close to uniform. Note that only a subset of question—26 out of 90—families is visualized in Figure 3 due to space limitation. Question examples of question families in which S1, S2, S3 and S4 are dominantly specialized are shown in Table 3.

Figure 4 shows the predictive distributions of MCL, CMCL and MCL-KD models with $k = 1$ on CLEVR dataset. MCL suffers from overconfidence issue that non-specialized models (*i.e.*, model 2, 3, 4, and 5) predict incorrect answers with high confidence, which leads to incorrect final decisions. Since the non-specialized models in CMCL are learned to generate uniform distribution and all models lose the opportunity to learn from a sufficient number of training examples, specialized model fails to predict the correct answer. However, MCL-KD predicts the correct answer since some non-specialized models are capable of predicting correct answers using knowledge distilled from the corresponding base models.

## 4.2   Image Classification

**Dataset**   Although our primary objective is to learn specialized models for VQA, our algorithm is easily applicable to any other tasks. Thus, we also evaluate our algorithm on image classification tasks using CIFAR-100 [18] dataset, and compare its performance with IE, MCL, and CMCL again.

Table 4: Classification accuracy (%) on CIFAR-100 with varying the number of specialized models ($k$) out of 5 models. The test accuracies are represented as top1/oracle. The numbers in red and blue denote the best and second-best algorithms for each classification model over $k$ in top1 accuracy, respectively. FS means feature sharing proposed in CMCL.

| Model name | ResNet-20 | | | VGGNet-17 | | |
| --- | --- | --- | --- | --- | --- | --- |
| K | 1 | 2 | 3 | 1 | 2 | 3 |
| Single | | 53.98 / - | | | 61.62 / - | |
| IE | | 64.60 / 79.95 | | | 68.43 / 81.07 | |
| MCL | 56.40 / 70.40 | 57.19 / 78.28 | 61.28 / 80.03 | 51.94 / 78.64 | 62.41 / 82.77 | 67.30 / 83.91 |
| CMCL | 48.67 / 62.56 | 55.16 / 71.69 | 58.13 / 75.01 | 57.27 / 62.16 | 64.37 / 74.41 | 67.44 / 78.90 |
| CMCL+FS | 56.30 / 70.39 | 61.20 / 77.24 | 63.78 / 79.84 | 60.85 / 66.05 | 66.34 / 75.59 | 67.94 / 80.36 |
| MCL-KD | **65.27** / 80.74 | **65.31** / 80.79 | **65.60** / 80.91 | **68.75** / 81.87 | **68.77** / 81.72 | **68.80** / 82.00 |
| MCL-KD+FS | **66.72** / 81.07 | **66.63** / 81.86 | **66.56** / 81.33 | **69.70** / 82.48 | **69.33** / 82.09 | **68.92** / 82.11 |

CIFAR-100 [18] dataset consists of 50,000 training and 10,000 test images with 100 image classes, where each image consists of $32 \times 32$ RGB pixels. This dataset has a significantly large number of classes compared to the ones tested in Lee *et al*. [19]—CIFAR-10 and SVHN, and has a limited number of training example per class. Therefore, performance improvement by model specialization is known to be challenging under the existing multiple choice learning framework.

**Experimental setup**   Following the original implementation of CMCL [19][2], we preprocess the images with global contrast normalization and ZCA whitening, and do not use any data augmentation. We employ two convolutional neural networks including VGGNet-17 [32] and ResNet-20 [11]. For all experiments, we use softmax classifier, and each model is optimized using stochastic gradient descent algorithm with Nesterov momentum. For this task, we also consider feature sharing, which stochastically shares the features among ensemble members. This trick is proposed in CMCL to improve classification performance by sharing general features and reducing data deficiency issue. As in [19], we share the non-linear activated features right before the first pooling layer, *i.e.*, the 2nd ReLU activations of VGGNet-17 and the 6th ReLU activations of ResNet-20.

**Results**   Table 4 presents the results of all models on CIFAR-100 dataset. Both MCL and CMCL fail to achieve competitive accuracy compared to IE regardless of $k$. It is surprising that their oracle accuracies are often worse than those of IE. This is because the number of training examples per class is only 500 in CIFAR-100, and the data deficiency problem drives MCL and CMCL to fail in specialization. Note that, although feature sharing of low-level representations turns out to be helpful to improve classification accuracy in all cases by addressing data deficiency issue, its benefit is not sufficient to improve accuracy in MCL and CMCL on CIFAR-100 dataset. However, by adopting knowledge distillation, our method consistently outperforms IE for all $k$'s by large margins with and without feature sharing.

## 5   Conclusion

We presented a novel and principled framework to learn specialized models for visual question answering. For the purpose, we formulate the problem with an ensemble of models, where each model is specialized dynamically on a subset of training examples within a multiple choice learning framework. By exploiting the idea of knowledge distillation, we first learn base models with the whole data and specialize models to predict ground-truth labels on assigned examples while preserving the representations of the base models on non-assigned ones. This method effectively addresses the data deficiency issues in multiple choice learning. We showed that our algorithm consistently outperforms all other methods including IE, MCL, CMCL in VQA and image classification. We believe that adaptively determining the number of models for specialization of each example would be an interesting future direction.

**Acknowledgments**

This work was partly supported by ICT R&D program of the MSIP/IITP grant [2016-0-00563; Research on Adaptive Machine Learning Technology Development for Intelligent Autonomous Digital Companion, 2017-0-01778; Development of Explainable Human-level Deep Machine Learning Inference Framework] and Kakao and Kakao Brain corporations.

## Footnotes

[1]https://github.com/hengyuan-hu/bottom-up-attention-vqa

[2]https://github.com/chhwang/cmcl

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
