[Reviews · NeurIPS 2018]

Reviewer 1



— This paper introduces an algorithm for learning specialized models in a multiple choice learning framework for visual question answering. — The authors make the observation that naively applying MCL to VQA leads to poor generalization because of data deficiency — each model looks at fewer training examples than a single model trained on the whole dataset — and more crucially, they miss out on compositional information. For example, one model might be specialized for ‘what color is the umbrella?’ and another for ‘how many people are wearing glasses?’ while at test time they question may be ‘what color are the glasses?’. — To work around this, the authors augment MCL with distillation (MCL-KD). Specifically, they train independently ensembled base VQA models on the entire dataset, and then while training using MCL, subset of models are trained using oracle assignments (as in usual MCL) while the rest are trained to imitate the base models’ activations. — The authors demonstrate impressive results on CLEVR and CIFAR-100 using the proposed approach, wherein they compare against MCL and Confident MCL. Strengths — The paper is very nicely written. It starts with a clear description of the problem, the observations made by the authors, and then the proposed solution — positioning it appropriately with respect to prior work — and then experiments. — Results overall, and especially on CIFAR are quite impressive. Given the small dataset, MCL and CMCL perform worse than independent ensembling, while MCL-KD performs better. Weaknesses — How does feature sharing affect performance in case of VQA? That might obviate the need for knowledge distillation to some extent as all models till a specific layer will now get to see the entire dataset, thus having access to compositional information. If there are complementary gains from FS and KD, do the authors have intuitions on what they are? — I have a small confusion in table 1. Maybe I’m misinterpreting something, but why are top-1 accuracies for MCL at k = 1 so low? Especially since these are initialized with IE base models. Evaluation Overall, this is good work, and I’m happy to recommend this for publication. The authors study an interesting and useful extension to MCL / CMCL. In terms of quantitative results, the proposed approach outperforms prior work on CLEVR and CIFAR, and qualitative results on CLEVR do show that some amount of specialization to question families is induced in these models through this training process.

Reviewer 2



Post-rebuttal Update ================= Given the authors response and the inclusion of VQAv2 experiments, I'm more convinced of the method's efficacy and recommend acceptance. ---------- Summary: ---------- This submission puts forth a knowledge-distillation based loss term to the Multiple Choice Learning framework of [9]. The propose approach is a somewhat minor (though impactful) deviation from the Confident Multiple Choice Learning approach of [18]. Specifically, the KL divergence to a uniform distribution for non-minimal loss models is targeted instead at the output distribution of some pretrained base model. This tweak seems to stabilize the performance of the ensemble as a whole -- leading to significantly improved top-1 accuracy, albeit at somewhat lower oracle performance when sufficient data is present. ---------- Clarity: ---------- Overall the paper is clearly written. I do have a couple of minor quesitons/concerns: - It is unclear how the question clusters were determined for CLEVR? - Some comments in the paper seem overly harsh to past methods (see L155 for an example). I'm sure these are oversights that sneak into submissions as deadlines loom, but it is worth restating some borrowed wisdom: "Existing approaches need not be bad for your method to be good." - The swap from accuracy in Table 1 to error rate in Table 2 is a little disruptive. ---------- Quality: ---------- Experimental Setting: The title and introduction of the paper argues quite well for the need for specialization in the context of visual question answering, but the experimental setting is less convincing. I've listed some concerns/questions below: - Why was CLEVR chosen as the evaluation setting for VQA? Does it have some property that regular VQA datasets don't? CLEVR is meant as an diagnostic dataset for relational reason and furthermore is essentially solved. As such, it is somewhat unclear whether ensemble improvements in this setting extend to real VQA datasets. I would like to see experiments on the VQA 2.0 dataset. - How was the balancing term \beta set? L155 says it was set empirically, but results on CLEVR are reported on the validation set. Was an additional set withheld from the training set to evaluate this parameter or was it tuned by hand for validation accuracy? I would like to see a chart of oracle and top-1 performance as \beta is varied. I'm conflicted on the CIFAR100 experiments. On one hand, it isn't clear how well they fit with the title / motivation of the paper as addressing a VQA problem. On the other, they do establish generality of the method and explore a very data sparse setting for MCL methods. ---------- Originality: ---------- As noted in the summary, the technical novelty in this submission is somewhat limited over prior work. While the contribution is simple, if it can be shown to robustly improve over existing approaches, then I consider its simplicity a virtue rather than a cause for alarm. ---------- Significance: ---------- If the requested results come back positive, I think the paper could be a significant work -- offering a simple approach to improving deep ensembles.

Reviewer 3



# Response to authors’ response # I appreciate the authors' effort to include (1) VQA2 (2) an MLP baseline (3) the result at k = 4. Overall, I like the paper’s idea and am satisfied with the response, although the improvement from the independent random ensemble (IE) to the proposed method (MCL-KD) becomes less significant at (1) and (2). The performance of a single model on VQA2 seems to be lower than that reported in [14]. The authors should check on this. Reviewer 2 and I have several similar concerns---the paper needs to work on real datasets and more algorithms to show its robustness. I strongly suggest the authors to include one more dataset and at least one more algorithm. Also, I would like to see analysis on models learned for VQA2---is each model specialized to a different question type? The paper could bring fruitful thinking and impact to the Visual QA community. I will maintain my score for paper acceptance. My specific response is in [] after each comment. ========================================================================= # Summary # This paper aims to learn an ensemble of specialized models for Visual QA---each model ideally will be dedicated to different question (or image) styles. The authors propose an objective with distillation on top of the “multiple choice learning MCL framework” to prevent (1) data deficiency on learning each specialized model and (2) forgetting general knowledge. The contribution is complementary and compatible to different Visual QA models (most of the Visual QA papers focus on developing new models), and thus could potentially benefit the whole community. # Strengths # S1. As mentioned above, the proposed objective is model-agnostic---any existing Visual QA model can potentially be applied to be the specialized model. This is particularly important in the literature of Visual QA---there have been too many papers (at least 30) on proposing different models but the performance gap is very limited and it is hard to tell what are the essential insights / concepts / components for Visual QA. The authors should emphasize this. S2. The paper is clearly written. The idea is clean and inspiring. The performance improvement compared to the existing MCL and CMCL objectives is significant and consistent. # Main weaknesses (comments) # W1. The paper only evaluates on a synthetic dataset (i.e., CLEVR). It will be important to evaluate on real datasets like VQA2 or Visual Genome to demonstrate the applicability. It will be interesting to see if each specialized model will be dedicated to different questions types (e.g., questions begin with why, what, how). [The authors do provide results on VQA2, but the improvement is largely reduced. Moreover, the single model performance on VQA2 is lower than reported in [14]. What is the “k” here? In the future, the authors can use the datasets by K. Kafle and C. Kanan, An Analysis of Visual Question Answering Algorithms, ICCV 2017, which has less bias issues.] W2. To support the strength, it will be important to evaluate on different Visual QA models beyond stacked attention network. A simple MLP baseline is a good starting point. See the following paper as an example to show its applicability. [The authors add the experiments. Although on MLP the gain becomes smaller, it still shows improvement. I would like to see more on this in the final version. Again, what is the “k” here?] Hexiang Hu, Wei-Lun Chao, and Fei Sha, Learning answer embeddings for visual question answering, CVPR 2018 It will be great if the authors can also discuss other papers that can potentially benefit the whole community such as Kushal Kafle and Christopher Kanan, Answer-Type Prediction for Visual Question Answering, CVPR 2016 W3. Besides the MCL framework, the authors should discuss other ways to learn an ensemble of models (e.g., mixture of experts) and the possibility (or difficulties) to apply them. [The authors promise to add the discussions.] # Minor weaknesses (comments) # W4. Eq. (6) seems not mention the case where k > 1. Should it be argmin for top k? Eq. (4) and (5) might be simplified by removing Z and \nu---\nu(Z(x)) is actually P(y|x). W5. From Table 1, it seems that the larger the k is, the higher the performance is. It would be important to evaluate on k = 4 to see if the performance will still increase or decrease to demonstrate the need of specialized models. [The authors show that the performance degrades when k= 4. Thanks.] W6. What is M for the image classification experiment? # Further comments # how to bridge the gap between the oracle and top 1 performance seems to be a promising direction for future work. I’m thinking if changing the decision rule in Eq. (8) by learning an attention p(m|x) on how confidence each model can help. Specifically, if p(m|x) can give higher weight to the model that gives the correct prediction, we can achieve the oracle performance.